# Group-based Motion Prediction for Navigation in Crowded Environments

**Allan Wang**
Robotics Institute
Carnegie Mellon University

**Christoforos Mavrogiannis**
Paul G. Allen School of Computer Science & Engineering
University of Washington

**Aaron Steinfeld**
Robotics Institute
Carnegie Mellon University

**Abstract:** We focus on the problem of planning the motion of a robot in a dynamic multiagent environment such as a pedestrian scene. Enabling the robot to navigate safely and in a socially compliant fashion in such scenes requires a representation that accounts for the unfolding multiagent dynamics. Existing approaches to this problem tend to employ microscopic models of motion prediction that reason about the individual behavior of other agents. While such models may achieve high tracking accuracy in trajectory prediction benchmarks, they often lack an understanding of the group structures unfolding in crowded scenes. Inspired by the Gestalt theory from psychology, we build a Model Predictive Control framework (G-MPC) that leverages group-based prediction for robot motion planning. We conduct an extensive simulation study involving a series of challenging navigation tasks in scenes extracted from two real-world pedestrian datasets. We illustrate that G-MPC enables a robot to achieve statistically significantly higher safety and lower number of group intrusions than a series of baselines featuring individual pedestrian motion prediction models. Finally, we show that G-MPC can handle noisy lidar-scan estimates without significant performance losses.

## 1 Introduction

Over the past three decades, there has been a vivid interest in the area of robot navigation in pedestrian environments [1, 2, 3, 4, 5]. Planning robot motion in such environments can be challenging due to the lack of rules regulating traffic, the close proximity of agents and the complex emerging multiagent interactions. Further, accounting for human safety and comfort as well as robot efficiency add to the complexity of the problem.

To address such specifications, a common [3, 4, 6, 7, 8] paradigm involves the integration of a behavior prediction model into a planning mechanism. Recent models tend to predict the individual interactions among agents to enable the robot to determine collision-free candidate paths [3, 4, 9]. While this paradigm is well-motivated, it tends to ignore the structure of interaction in such environments. Often, the motion of pedestrians is coupled as a result of social grouping. Further, the motion of multiple agents can often be *effectively* grouped as a result of similarity in motion characteristics. Lacking a mechanism for understanding the emergence of this structure, the robot motion generation mechanism may yield unsafe or uncomfortable paths for human bystanders, often violating the space of social groups.

Motivated by such observations, we draw inspiration from human navigation to propose the use of group-based prediction for planning in crowd navigation domains. We argue that humans do not employ detailed individual trajectory prediction mechanisms. In fact, our motion prediction capabilities are short-term and do not scale with the number of agents. We do however employ effective grouping techniques that enable us to discover safe and efficient paths among motions of crowd networks. This anecdotal observation is aligned with gestalt theory from psychology [10] which suggests that organisms tend to perceive and process *formations of entities*, rather than individual

5th Conference on Robot Learning (CoRL 2021), London, UK.

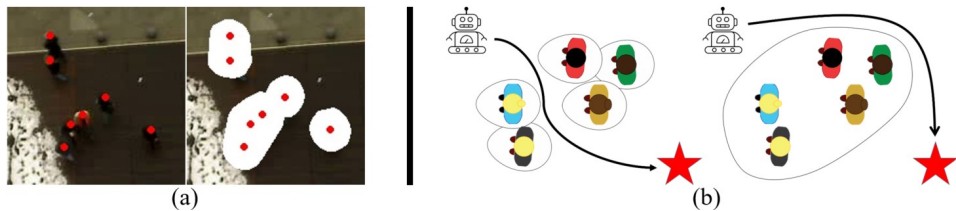

Figure 1: Based on a representation of social grouping [13], we build a group behavior prediction model to empower a robot to perform safe and socially compliant navigation in crowded spaces. (a) Example of our representation overlayed on top of a scene from a real-world dataset [14]. (b) A model predictive controller equipped with our prediction model is able to navigate around the group socially (right) as opposed to the baseline that cuts through the group (left).

components. Such techniques have recently led to advances in computer vision [11] and computational photography [12]. Similarly, we envision that a robot could reason the formation of effective groups in a crowded environment and react to their motion as an effective way to navigate safely.

In this paper, we propose a group-based representation coupled with a prediction model based on the group-space approximation model of Wang and Steinfeld [13]. This model groups a crowd into sets of agents with similar motion characteristics and draws geometric enclosures around them, given observation of their states. The prediction module then predicts future states of these enclosures. We conduct an extensive empirical evaluation over 5 different human datasets [14, 15], each with a flow following and a crossing scenario. We further conduct a same set of evaluations with agents powered by ORCA [16] that share the start and end locations in the datasets. Last but not least, we conducted evaluation given inputs in the form of simulated laser scans, from which pedestrians are only partially observable or even completely occluded. We compare the performance of our group-based formulation against three individual reasoning baselines: a) a reactive baseline with no prediction; b) a constant velocity prediction baseline; c) one based on individual S-GAN trajectory predictions [17]. We present statistically significant evidence suggesting that agents powered by our formulation produce safer and more socially compliant behavior and are potentially able to handle imperfect state estimates.

## 2 Related Work

Over the recent years, a considerable amount of research has been placed to the problem of robot navigation in crowded pedestrian environments [3, 4, 8, 18, 19, 20, 21, 22, 23]. Such environments often comprise groups of pedestrians, navigating as coherent entities. Šochman and Hogg [24] suggests that 50-70% of pedestrians walk in groups. Many works exist in group detection. One popular area in such domain is static group detection, often leveraging F-formation theories [25]. However, dynamic groups often dominate pedestrian-rich environments and exhibit different spatial behavior [26]. Among dynamic group detection, one common approach is to treat grouping as a probabilistic process where groups are a reflection of close probabilistic association of pedestrian trajectories [27, 28, 29, 30, 31]. Others use graph models to build inter-pedestrian relationships with strong graphical connections indicating groups [32, 33]. The social force model [34] also inspires Mazzon et al. [35], Šochman and Hogg [24] to develop features that indicate groups. Clustering is another common group of technique to group pedestrians with similar features into groups [36, 37, 38, 39]. For our formulation, it is sufficient to employ a simple clustering-based grouping method proposed by Chatterjee and Steinfeld [39]. Other grouping methods will simply result in different group membership assignments.

Applications on groups often focus on a specific behavior aspect. For example, one focus in this area is how a robot should behave as part of the group formation [40]. On dyad groups involving a single human and a robot, some researchers examined socially appropriate following behavior [41, 42, 43, 44] and guiding behavior [45, 46, 47]. In works that do not include robots as part of pedestrian groups, some research teams studied how a robot should guide a group of pedestrians [48, 49, 50]. From navigation perspective, Yang and Peters [26] leverage groups as obstacles, but their group space only involves occasional O-space modeling from F-formation theories. Without the engineered occurrence of O-space, their representation reduces to one of our baselines. Katyal et al. [51] introduce an additional cost term that leverages robot's distance to the closest group in

a reinforcement learning framework. They model groups using convex hulls directly generated from pedestrian coordinates instead of taking personal spaces into consideration. In our work, we additionally explore the capabilities of groups in handling imperfect sensor inputs. While our focus is on analysing the benefits of groups, our group based formulation can be easily incorporated into the work of Katyal et al. [51]'s framework.

# 3  Problem Statement

Consider a robot navigating in a workspace $\mathcal{W} \subseteq \mathbb{R}^2$ amongst $n$ other dynamic agents. Denote by $s \in \mathcal{W}$ the state of the robot and by $s^i \in \mathcal{W}$ the state of agent $i \in \mathcal{N} = \{1, \ldots, n\}$. The robot is navigating from a state $s_0$ towards a destination $s_T$ by executing a policy $\pi : \mathcal{W}^{n+1} \times \mathcal{U} \to \mathcal{U}$ that maps the assumed fully observable world state $\boldsymbol{S} = s \cup_{i=1:n} s^i$ to a control action $u \in \mathcal{U}$, drawn from a space of controls $\mathcal{U} \subseteq \mathbb{R}^2$. We assume that the robot is not aware of agents' destinations $s_T^i$ or policies $\pi_i : \mathcal{W}^{n+1} \times \mathcal{U}^i \to \mathcal{U}^i, i \in \mathcal{N}$. In this paper, our goal is to design a policy $\pi$ that enables the robot to navigate from $s_0$ to $s_T$ safely in a socially compliant fashion.

# 4  Group-based Prediction

We introduce a group representation building on prior work [13] and a model for group-based prediction that is amenable for use in decentralized multiagent navigation.

## 4.1  Group Representation

Define as $\theta^i \in [0, 2\pi)$ the orientation of agent $i \in \mathcal{N}$ which is assumed to be aligned with the direction of its velocity $u^i$, extracted via finite differencing of its position over a timestep $dt$ and denote by $v^i = ||u^i|| \in \mathbb{R}^+$ its speed. We define an augmented state for agent $i$ as $q^i = (s^i, \theta^i, v^i)$.

We treat a social group as a set of agents who are in close proximity and share similar motion characteristics. Assume that a set of $J$ groups, $\mathcal{J} = \{1, \ldots, J\}$ navigate in a scene. Define by $g^i \in \mathcal{J}$ a label indicating the group membership of agent $i$. We then define a group $j \in \mathcal{J}$ as a set $G^j = \{i \in \mathcal{N} \mid g^i = j\}$ and collect the set of all groups in a scene into a set $\boldsymbol{G} = \{G^j \mid j \in \mathcal{J}\}$.

**Extracting Group Membership.** We define the combined augmented state of all agents as $\boldsymbol{q} = \cup_{i=1:n} q^i$. To obtain group memberships for a set of agents $\mathcal{N}$, we apply the Density-Based Spatial Clustering of Applications with Noise algorithm (DBSCAN) [52] on agent states:

$$\boldsymbol{G} \longleftarrow \texttt{DBSCAN}(\boldsymbol{q} \mid \epsilon_s, \epsilon_\theta, \epsilon_v) \tag{1}$$

Where $\epsilon_s, \epsilon_\theta, \epsilon_v$ are respectively threshold values on agent distances, orientation and speeds.

**Extracting the Social Group Space.** For each group $G^j$, $j \in \mathcal{J}$, we define a *social group space* as a geometric enclosure $\mathcal{G}^j$ around agents of the group. For each agent $i \in G^j$, we define a personal space $\mathcal{P}^i$ as a two-dimensional asymmetric Gaussian based on the model introduced by Kirby [53]. Refer to Appendix A for detailed descriptions.

Given the personal spaces $\mathcal{P}^i$, $i \in G^j$, of all agents in a group $j$, we extract the social group space of the whole group as a convex hull:

$$\mathcal{G}^j = \text{Convexhull}(\{\mathcal{P}^i \mid i \in G^j\}). \tag{2}$$

The shape described by $\mathcal{G}^j$ represents an obstacle space representation of a group containing agents in close proximity with similar motion characteristics. For convenience, let us collect the spaces of all groups in a scene into a set $\boldsymbol{\mathcal{G}} = \{\mathcal{G}^j \mid j \in \mathcal{J}\}$.

## 4.2  Group Space Prediction Oracle

Based on the group-space representation of Sec. 4.1, we describe a prediction oracle that outputs an estimate of the future spaces occupied by a set of groups $\boldsymbol{\mathcal{G}}_{t:t_f}$ up to a time $t_f = t + f$, where $f$ is a future horizon given a past sequence of group spaces $\boldsymbol{\mathcal{G}}_{t_h:t}$ from time $t_h = t - h$ where $h$ is a window of past observations:

$$\boldsymbol{\mathcal{G}}_{t:t_f} \leftarrow \mathcal{O}(\boldsymbol{\mathcal{G}}_{t_h:t}) = \cup_{j=1:J} \mathcal{O}_j(\mathcal{G}_{t_h:t}^j), \tag{3}$$

Table 1: Prediction Model Performance

| | Metric | ETH | HOTEL | ZARA1 | ZARA2 | UNIV |
|---|---|---|---|---|---|---|
| Baseline | mIoU (%) | 83.52 | 90.37 | 88.04 | 89.30 | 85.32 |
| | fIoU (%) | 76.32 | 85.38 | 82.14 | 83.88 | 77.24 |
| Encoder-decoder Network | mIoU (%) | 86.66 | 92.10 | 89.97 | 90.94 | 87.52 |
| | fIoU (%) | 78.64 | 86.83 | 83.77 | 85.09 | 78.55 |

where $\mathcal{O}_j$ is a model generating a group space prediction for group $G^j$. Refer to Appendix B for detailed description of partial input handling.

We implement the oracle $\mathcal{O}_j$ of eq. (3) using a simple encoder-decoder network. The encoder follows the 3D convolutional architecture in [54] whereas the decoder mirrors the model layout of the encoder. The encoder-decoder network takes as input a sequence[1] $\mathcal{G}_{t_h:t}$ and outputs a sequence $\mathcal{G}_{t+1:t_f}$ which we pass through a sigmoid layer. We supervise the encoder-decoder network's output using the binary cross entropy loss.

We verified the effectiveness of our encoder-decoder network on the 5 scenes of our experiments by conducting a cross-validation comparison against a baseline. The baseline predicts the future shapes by linearly translating the last social group shape using its geometric center velocity. We use Intersection over Union (IoU) as our metric. Between the ground truths and the predictions, this metric divides the number of overlapped pixels by the number of pixels occupied by either one of them. As shown in Table 1, our encoder-decoder network outperforms the baseline.

## 5 Model Predictive Control with Group-based Prediction

We describe G-MPC, a model predictive control (MPC) framework for navigation in multiagent environments that leverages the group-based prediction oracle of Sec. 4.

We describe our group-prediction informed MPC, or G-MPC. At planning time $t$, given a (possibly partial) augmented world state history $\boldsymbol{Q}_{t_{\hat{h}}:t}$, we first extract a sequence of group spaces $\mathcal{G}_{t_{\hat{h}}:t}$ based on the method of Sec. 4.1. Given these, the robot computes an optimal control trajectory $\boldsymbol{u}^* = u_{1:K}^*$ of length $K$ by solving the following optimization problem:

$$(\boldsymbol{s}^*, \boldsymbol{u}^*) = \arg\min_{u_{1:K}} \sum_{k=1:}^{K} \gamma^k J(s_{k+1}, \boldsymbol{\mathcal{G}}_{k+1}, s_T) \tag{4}$$

$$s.t.\ \boldsymbol{\mathcal{G}}_{2-h:1} \leftarrow \boldsymbol{\mathcal{G}}_{t_h:t} \tag{5}$$

$$s_1 \leftarrow s_t \tag{6}$$

$$\boldsymbol{\mathcal{G}}_{k+1:k_f} = \mathcal{O}(\boldsymbol{\mathcal{G}}_{k_h:k}) \tag{7}$$

$$u_k \in \mathcal{U} \tag{8}$$

$$s_{k+1} = s_k + u_k \cdot dt, \tag{9}$$

where $\gamma$ is the discount factor and $J$ represents a cost function, eq. (5) initializes the group space history ($k = 2 - h$ is the timestep displaced a horizon $h$ in the past from the first MPC-internal timestep $k = 1$), eq. (6) initializes the robot state to the current robot state $s_t$, eq. (7) is an update rule recursively generating a predicted future group sequence up to timestep $k_f = k + f$ given history from time $k_h = k - h$ up to time $k$, $\mathcal{O}$ represents a group-space prediction oracle based on Sec. 4, and eq. (9) is the robot state transition assuming a fixed time parametrization of step size $dt$.

We employ a weighted sum of costs $J_g$ and $J_d$, penalizing respectively distance to the robot's goal and proximity to groups:

$$J(s_k, \boldsymbol{\mathcal{G}}_k, s_T) = \lambda J_g(s_k, s_T) + (1 - \lambda) J_d(s_k, \boldsymbol{\mathcal{G}}_k), \tag{10}$$

where $\lambda$ is a weight representing the balance between the two costs and

$$J_g(s_k) = \begin{cases} 0, & \text{if } s_k \in \boldsymbol{\mathcal{G}}_k \\ ||s_{k-1} - s_T||, & \text{else,} \end{cases} \tag{11}$$

---

[1]The oracle input sequence is first converted into image-space coordinates using the homography matrix of the scene. We also preprocess inputs to have normalized scale and group positions. The autoencoder output is converted back into Cartesian coordinates using the inverse homography transform.

Table 2: Number of trials per task and scene.

| Task | ETH | HOTEL | ZARA1 | ZARA2 | UNIV |
|------|-----|-------|-------|-------|------|
| Flow | 58 | 43 | 25 | 127 | 106 |
| Cross | 58 | 44 | 28 | 129 | 114 |

penalizes a rollout according to the distance of the last collision-free waypoint to the robot's goal. Further, we define $J_d$ as:

$$J_d(s_k, \boldsymbol{\mathcal{G}}_k) = \exp(-\mathcal{D}(s_{k+1}, \boldsymbol{\mathcal{G}}_k)), \tag{12}$$

where

$$\mathcal{D}(s_k, \boldsymbol{\mathcal{G}}_k) = \begin{cases} \min_{j \in \mathcal{J}} D\left(s_k - \mathcal{G}_k^j\right), & \text{if } s_k \notin \mathcal{G}_k^j \\ -\min_{j \in \mathcal{J}} D\left(s_k - \mathcal{G}_k^j\right), & \text{else,} \end{cases} \tag{13}$$

where $D(s_k - \mathcal{G}_k^j)$ returns the minimum distance between the robot state and the space occupied by group $j$ at time $k$. Using $D$, function $\mathcal{D}$ computes the minimum distance to any group for a given time. In most cases, the robot lies outside of groups, i.e., $s_k \notin \mathcal{G}_k^j$ –therefore, the cost $J_d$ tries to maximize the distance $\mathcal{D}$. Sometimes, the robot might end up entering the group space $\boldsymbol{\mathcal{G}}$ –in those cases, $J_d$ tries to minimize $\mathcal{D}$, to steer the robot towards the direction of quickest escape from the group. In case that the robot is inside a group to begin with, we shrink the group sizes in Sec. 4.1 until the robot is outside the groups again.

To solve eq. (4), we search over a finite set $\boldsymbol{\mathcal{U}}$ of control trajectories of horizon $K$. With the assumption that the robot is holonomic and is not under any kinematic constraints, we use a set of $R$ control rollouts $\boldsymbol{\mathcal{U}} = \{\boldsymbol{u}^1, ..., \boldsymbol{u}^R\}$ with three levels of tangential speeds and a set of turning speed, i.e.,

$$u_{1:K}^r = (v\cos\psi, v\sin\psi, \omega),\ \psi = \frac{2\pi r}{R}, v \in \left\{\frac{1}{3}v_{max}, \frac{2}{3}v_{max}, v_{max}\right\}, \omega \in \left\{0, \pm\frac{\pi}{2}\right\} \tag{14}$$

To ensure compatibility between our group-based prediction model and our MPC formulation, we set the control rollout time horizon to be the prediction model's prediction horizon, or $K = f$.

## 6  Evaluation

We evaluate our framework through a simulation study in which the robot performs a navigation task (a transition between two points) within a crowds of dynamic agents in a set of scenes.

### 6.1  Experimental Setup

We consider a set of realistic pedestrian scenes, drawn from the ETH [14] (ETH and HOTEL scenes) and UCY [15] (ZARA1, ZARA2 and UNIVERSITY scenes) datasets, which often serve as benchmarking testbeds in the motion prediction and social navigation literature [17, 55, 56, 57]. In each scene, we define two navigation tasks (see Fig. 2): *Flow*: in which the robot navigates along the crowd flow and *Cross* in which the robot intersects vertically with the traffic flow. For each task, we generate a set of trials by segmenting the scene recording into blocks involving challenging interactions. We define a challenging interaction to be a segment involving at least 5 pedestrians inside the test region drawn in black in Fig. 2. This process provided us with a distribution of trials as shown in table Table 2. Across all trials, we keep the robot's maximum speed at $1.75m/s$.

We consider two experimental conditions: an *Offline* and an *Online* one. In the *Offline* one, the robot navigates among a crowd moving according to a recording of a human crowd. Under this condition, pedestrians act as dynamic obstacles that do not react to the robot, a situation which could arise in cases where robots are of shorter size and could thus be easily missed by navigating pedestrians. In the *Online* one, the robot navigates among a crowd[2] moving by running ORCA [16], a policy that is frequently used as a simulation engine for benchmarking in the social navigation literature [8, 57, 58].

To investigate the value of G-MPC, we develop three variants of it. **group-pred** is a G-MPC in which the encoder-decoder network has a history $h = 8$ and a horizon $f = 8$. **group-nopred** is

---

[2]For consistency, the agents in the crowd start and end at the same spots as the agents in the recorded crowd from the Offline condition.

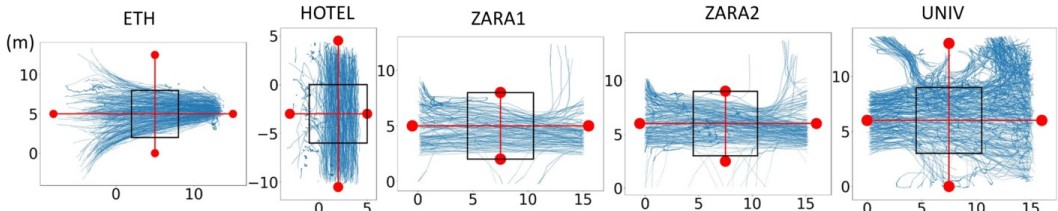

Figure 2: Trajectories of all pedestrians in the datasets. The red dots represent the task start and end locations. The red lines represent the task paths. The black box represents the test region to check for non-trivial tasks.

a variant that features no prediction at all –it just reacts to observed groups at every timesteps and it is equivalent to the framework of Yang and Peters [26]. Finally, **laser-group-pred** is identical to **group-pred** but instead of using ground-truth pose information, it takes as input noisy lidar scan readings. We simulate this by modeling pedestrians as $1m$-diameter circles and lidar scans as rays projecting from the robot. We refer to the spec sheet of a SICK LMS511 2D lidar for simulation parameters. We further inject noise into the readings according to the spec sheet. Under this simulation, pedestrians may only be partially observable or even completely occluded from the robot.

We compare the performance of these policies against a set of MPC variants using mechanisms for individual motion prediction. **ped-nopred** is a vanilla MPC that reacts to the current states of other agents without making predictions about their future states. **ped-linear** is a vanilla MPC that estimates future states of agents by propagating agents' current velocities forward. This baseline is motivated by recent work showing that constant-velocity models yield competitive performance in pedestrian motion prediction tasks [59]. Finally, **ped-sgan** is an MPC that uses S-GAN [17] to extract a sequence of future state predictions for agents based on inputs of their past states. We selected S-GAN because it is a recent highly performing model. To ensure a fair comparison, all the MPC policy variants are integrated with the same MPC controller evaluated at $dt = 0.1$.

We measure the performance of the policies with respect to four different metrics: a) *Success rate*, defined as the ratio of successful trials over total number of trials; b) *Comfort*, defined as the ratio of trials in which the robot does not enter any social group space over the total number of trials; c) *Minimum distance to pedestrians*, defined as the smallest distance between the robot and any agent per trial; d) *Path length*, defined as the total distance traversed by the robot in a trial.

To track the performance of G-MPC, we design hypotheses targeting aspects of safety and group space violation which we investigate under both experimental conditions, i.e., offline and online:

**H1**: To explore the benefits of group based representations alone, we hypothesize that **group-nopred** is safer than **ped-nopred** while achieving similar success rates but worse efficiency.

**H2**: To explore the full benefit of group based formulation, we hypothesize that **group-pred** is safer than **ped-linear** and **ped-sgan** while achieving similar success rates but worse efficiency.

**H3**: To explore how our formulation handles imperfect inputs, we hypothesize that **laser-group-pred** achieves similar safety to **group-pred** while achieving similar success rate and efficiency.

**H4**: To check that our formulation is socially compliant, we hypothesize that **group-nopred**, **group-pred** and **laser-group-pred** violate agents' group space less often than the baselines.

### 6.2 Results

**Quantitative Analysis.** Fig. 3 and Fig. 4 contain bar charts representing the performance of G-MPC compared with its baselines under Offline and Online settings respectively. Bars indicate means, errorbars indicate standard deviations, "F" and "C" are flow and cross scenarios respectively, and the number of asterisks indicates increasing significance levels: $\alpha = 0.05, 0.01, 0.001$ according to two-sided Mann-Whitney U-tests.

**H1**: We can see from both Fig. 3 and Fig. 4 that G-MPC achieves statistically significantly larger minimum distances to pedestrians across all scenarios, often with $p < 0.001$. This illustrates that the group representation is in itself capable of upgrading a simple MPC with no prediction. As expected, we observe that the price G-MPC pays for that is a larger average path length. We also see that success rates are comparable. Overall, we conclude that H1 holds.

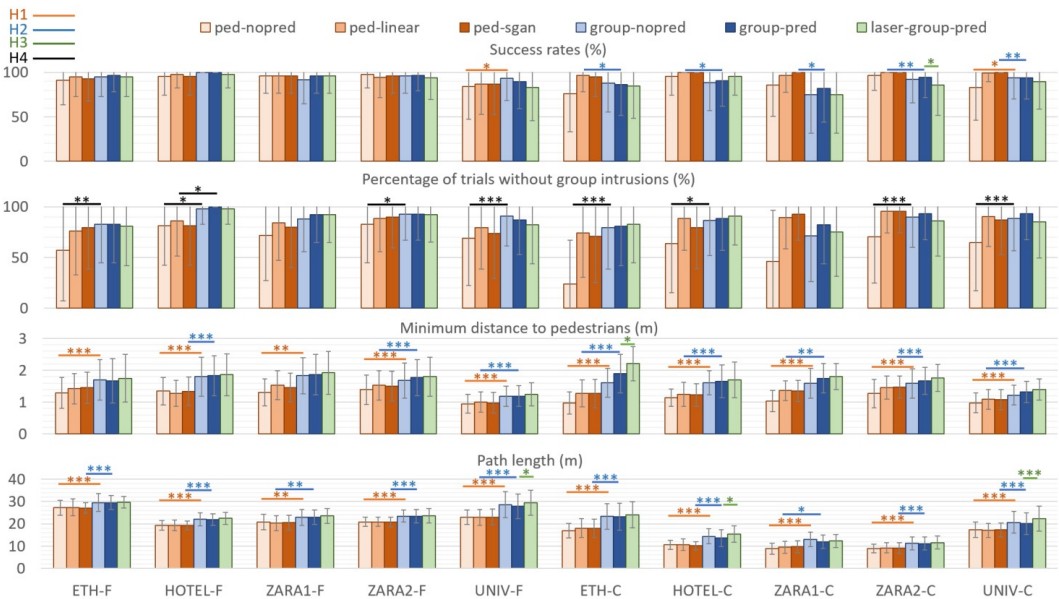

Figure 3: Performance per scene under the *Offline* condition. Horizontal lines indicate statistically significant results corresponding to different to hypotheses.

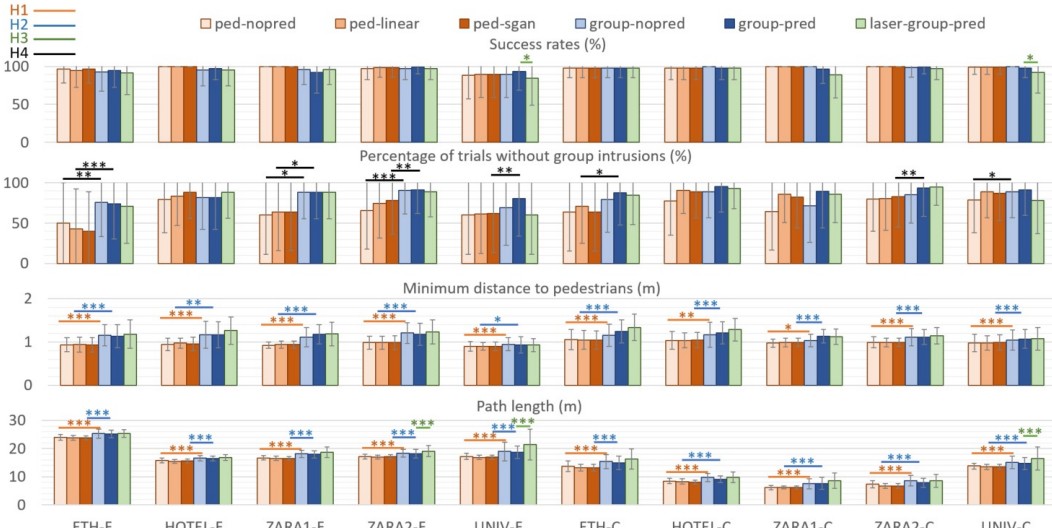

Figure 4: Performance per scene under the *Online* condition (simulated pedestrians powered by ORCA [16]). Horizontal lines indicate statistically significant results corresponding to different to hypotheses.

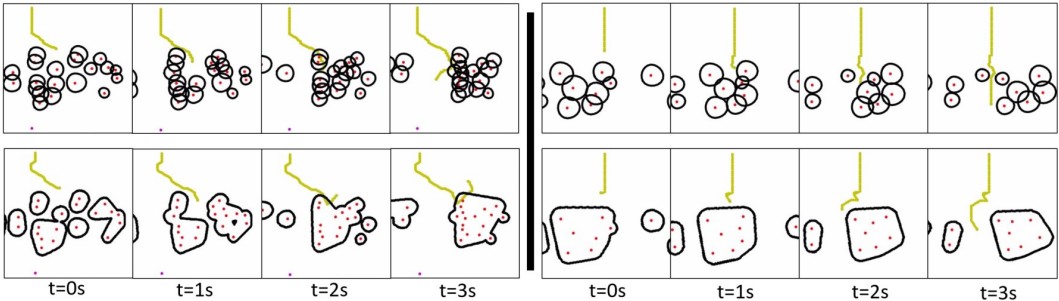

Figure 5: Qualitative performance difference between approaches leveraging pedestrian-based (top) and group-based (bottom) representations. Left: non-reactive agents. Right: reactive agents. In all cases, the robot starts from the top and attempts to navigate downward. The green line shows the robot's traversed trajectory.

**H2**: When future state predictions are considered, G-MPC obtains statistically significant results in most scenes supporting its attributes of being safer at the cost of worse efficiency. Thus H2 is partially confirmed. In offline scenarios, G-MPC has lower success rates in crossing scenarios. Upon closer inspection, most failure cases are due to timeouts from G-MPC's conservative behavior. However, in online scenarios where pedestrians react to the robot, G-MPC achieves high success rates. In real-world situations, to cross dense traffic, the robot needs to plan its actions with expectations of reactive pedestrians. Otherwise, the robot will most probably run into *the freezing robot problem* [4].

**H3**: Group-based representations have the potential to robustly account for imperfect state-estimates. Overall, we observe that with simulated imperfect states, G-MPC does not perform statistically significantly worse in terms of safety, but in dense crowds of the UNIV scenes it has worse efficiency and worse success rates in online cases. This shows that H3 holds in terms of safety and, in moderately dense human crowds, holds in terms of efficiency. Future work on better group representation is needed to achieve better efficiency in high-density human crowds given imperfect states.

**H4**: From Fig. 3 and Fig. 4, we can see that G-MPC often has fewer group-space intrusions than its baselines. While this relationship is not always statistically significant, we do see a general trend of the group-based approaches to respect group spaces more often than individual ones. Thus, we conclude that H4 is partially confirmed.

We additionally observe a general trend that **group-pred** is better than **group-nopred** in terms of higher success rates, lower chances of group intrusions, longer minimum distances to pedestrians and shorter path lengths. This shows that our group prediction model offers benefits to the robot's navigation. However, in a few scenarios **group-nopred** performs better. We largely attribute this to the finite inaccuracies of future group predictions and the freezing robot problem that accompanies the robot's more conservative behavior in **group-pred** than in **group-nopred**.

**Qualitative Analysis.** Qualitatively, it is a more common occurrence for regular MPC to perform aggressive and socially inappropriate maneuvers than G-MPC. As shown in the two examples in Fig. 5 executed by **ped-sgan** and **group-pred** agents, we can see that in offline conditions, the MPC agent aggressively cuts in front of the two pedestrians to the left before proceeding headlong into the cluster of pedestrians, only managing to avoid the deadlock by escaping through the narrow gap that opens up. While for G-MPC, it tracks the movements of the two pedestrian groups coming from the left. When the two pedestrian groups merge, the agent turns around and reevaluates its approach to cross. In the online condition, we observe that the MPC agent cuts through a pedestrian group to reach the other side, forcing a member of the group to stop and yield as indicated by the pedestrian's shrinking personal space, which is proportional to its speed. In the same situation, the G-MPC agent chooses to circumvent behind the social group.

# 7  Conclusion

We introduced a methodology for generating group-based representations and predicting their future states. Through an extensive evaluation over the flow and crossing scenarios drawn from 10 different real-world scenes from 2 different human datasets with both reactive and non-reactive agents, we demonstrate the value of group-based prediction in enabling safe and socially compliant navigation. Through experimentation with simulated laser scans, our model displays promising potential to process noisy sensor inputs without much performance downgrade.

Several improvements to our framework are possible. For example, we could incorporate state-of-the-art oracles in the form of advanced video prediction models [60] or incorporate inter-group interaction modeling. Additional considerations such as the set of rollouts or the cost functions could possibly increase performance. We could also integrate our prediction model into alternative control frameworks such as reinforcement learning policies.

Finally, we plan on validating our findings on a real-world robot to fully test the capability of G-MPC to handle noisy sensor inputs. We also plan to investigate ways to improve computation time to enhance our approach's real-world applicability. These include simplifying group representation geometry and predicting future group states in metric space instead of in image space.

## Acknowledgment

This work was funded by grant (IIS-1734361) from the National Science Foundation and Honda Research Institute USA.

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
