# OpenReview forum: "Group-based Motion Prediction for Navigation in Crowded Environments"
_robot-learning.org/CoRL/2021/Conference — CoRL2021 Oral_

### Official Review · Reviewer_rGtm · 2021-07-19

**Originality:** Very Good
**Technical Quality:** Excellent
**Clarity Of Presentation:** Good
**Impact:** 3

**Recommendation:**

Weak Accept: I recommend accepting the paper, but will not argue for my recommendation if the majority of other reviewers have a different opinion.

**Summary:**

This paper investigates motion planning among pedestrians, particularly focusing on the fact that pedestrians can travel in groups that have different dynamics/interactions than individual pedestrians. The technical contributions include a method for grouping/representing groups of pedestrians, an autoencoder to learn the group dynamics from a dataset, and an MPC formulation to plan collision-free paths with respect to the groups. The results include comparisons of various algorithms on fixed datasets and with simulations where agents follow ORCA policies.

**Issues:**

n/a

**Reviewer Expertise:**

Excellent: Expert knowledge on the topic of the paper

**Strengths And Weaknesses:**

Strengths:
- Groups are an important component of social robot navigation but most of the recent works on this topic focus on modeling individual pedestrians
- The paper is well organized, enjoyable to read, and the problem is well-motivated and explained.
- The approach of learning a group dynamics model and folding that into an MPC is interesting and makes sense, as MPC comes with collision guarantees, and it would be difficult to specify a group dynamics model without learning from data

Weaknesses:
- For groups, hardware experiments seem even more important than other social navigation problems, since the group dynamics are difficult to model and thus simulate. However, the authors designed insightful experiments in simulation and the challenges of running physical experiments (particularly with pedestrian groups) are totally understood here. Are there other ways to simulate group behaviors, other than ORCA (which is inherently single-agent) and playing back pre-recorded motions?
- The figures could be improved. I don't really understand Fig 5. Figs 3 and 4 are hard to see, especially since the tiny differences between algorithms are scrunched (e.g. for success rate, would it help to show failure rate instead?).

**Summary Of Recommendation:**

The paper lays out a principled way for group dynamics to be adopted into social navigation algorithms. Looking ahead, real-world testing of the idea will be important to determine the long-term impact of this type of modeling, though it seems like a promising direction.

---

> ### Author Response · Authors · 2021-08-28
> **Author Response to Reviewer rGtm**
>
> We greatly appreciate the thoughtful and constructive feedback provided by the reviewer. We are excited to hear your opinion that our presentation is clear and that our method is novel and intuitive. These appraisals also seemed to be shared by other reviewers. We are also very grateful for your suggestions to further improve the paper. We will address your major concern below.
>
> **Hardware experiments.** We would like to thank the reviewer for understanding the challenges we face running physical experiments. Due to special unforeseen circumstances, we were unable to perform analysis in the real world. Performing this type of analysis in the real world will require participants walking in groups and the study will likely involve a certain population density to take place. Unfortunately, this is difficult to accomplish reliably in our current locations due to COVID protocols. However, we did try our best to perform an in-depth empirical analysis in simulation, which we see as the first step towards a real-world deployment in the near future.
>
> Our offline dataset-based simulator contains real-world information of pedestrian grouping behavior. And the ETH and UCY datasets that we have been working with contain complex grouping behavior such as splitting, merging and collision avoidance as groups. The downside of this approach is that the pedestrians will not react to the robot. We would like to point out that this assumption can be a valid assumption within certain contexts, such as when the robot has a small, unnoticeable profile; when the surrounding pedestrians are in a hurry and do not wish to give way to the robot; or when the surrounding pedestrians are vulnerable (childrens, the elderly, pedestrians with disabilities etc.).
>
> Our online simulator compliments our offline simulator by adding a reactive component to the pedestrians. The downside is that the pedestrians are now artificially powered by ORCA. First, we still adopted the start and goal locations of the pedestrians as they appear and disappear in the datasets. This means that groups that first appear in the simulation are real-world groups. Second, despite being inherently individual-driven, ORCA has been used in other popular social navigation model evaluations [8, 57, 58]. The question is now are there other models that better simulate group behavior than ORCA? To the best of our knowledge, we believe that existing literature on computationally modeling pedestrian grouping behavior is scarce. Several existing social navigation approaches outperforms ORCA [4,8,19], but they are not focused on interacting with pedestrians as groups. We also mentioned in our literature review several works that focus on robot interactions with pedestrian groups, but they are not generic and only focus on specific types of interactions such as following and guiding.
>
> **Figures.** Thank you for your suggestions regarding our figures. We have added additional descriptions in the caption of figure 5 to aid understanding. For figures 3 and 4,  We can see that the bars are hard to see, especially for success rates and percentages of trials without group intrusions. Our major concern here is that because we are conducting statistical analysis, we believe it is essential to show the error bars. We have experimented with showing failure rates instead. What we discovered is that because the error bars are long, the differences between the bars are still scrunched. We do provide detailed numerical values of these bars in the appendix section (inside the supplementary materials folder).

---

> > ### Comment · Reviewer_rGtm · 2021-09-03
> > **revision**
> >
> > Thanks to the authors for the explanations - I am generally in agreement and my scores remain unchanged. Looking forward to future work, if hardware experiments continue to be impractical, the authors might consider developing new methods for simulating group dynamics in a more realistic manner.

---

### Official Review · Reviewer_UWWw · 2021-07-23

**Originality:** Very Good
**Technical Quality:** Excellent
**Clarity Of Presentation:** Very Good
**Impact:** 4

**Recommendation:**

Strong Accept: I recommend accepting the paper and will argue for my recommendation even if other reviewers hold a different opinion.

**Summary:**

The authors approach the problem of planning a robot trajectory through a dense pedestrian crowd. They cluster pedestrians into groups, and consider the combined personal space of such groups. Using past group-space image masks, future group-space masks are predicted by a 3D Convolutional Neural Network (CNN). A receding-horizon Model Predictive Control (MPC) approach is used to navigate the robot through the crowd.

**Issues:**

- The group space prediction oracle is repeatedly referred to as an "autoencoder". As the model predicts the *future* from the *past*, the input is not the output: therefore the network is not strictly an *auto-* encoder. Consider changing the language, for example to "encoder-decoder framework".
- Footnote 1 mentions a homographic transform between "cartesian" coordinates and "image-space" coordinates. The purpose and details of this transform could be elaborated in the supplementary details.
- (optional, stylistic) There are no less than 5 variables associated with the letter 'g' in various fonts, 4 of them capital 'G'. It might aid understanding if the distinction between group *membership* and group *space* was clearer.
- (note) This reviewer knows of only one other work with this macroscopic space-time-image representation of pedestrian crowds that the authors should consider as part of the related work: “Dynamic Crowd Prediction for Social Navigation”, S. H. Kiss, K. Katuwandeniya, A. Alempijevic, and T. Vidal-Calleja, published to ICRA 2021.

**Reviewer Expertise:**

Excellent: Expert knowledge on the topic of the paper

**Strengths And Weaknesses:**

Strengths

- Considering pedestrian crowds as larger groups than individuals is good insight, this research should encourage others in the field to consider "macroscopic" models.
- The evaluation of method work is significant, testing on a variety of public datasets. They also propose a method of testing via an interactive "Online" approach using ORCA; such a testing environment is very important when modelling interactive environments. In 4 cases, they prove their results are statistically significant.

Weaknesses

- From eq. (3), the learned model operates on each group individually. As such, there is no way for the model to capture inter-group interactions. This addition could be an interesting extension to this work.



**Summary Of Recommendation:**

The authors take an unconventional "macroscopic" approach to crowd prediction, and prove it is superior to state-of-the "microscopic" methods for online navigation. The authors clearly explain their approach, and back it up with an extensive evaluation section.

---

> ### Author Response · Authors · 2021-08-28
> **Author Response to Reviewer UWWw**
>
> We greatly appreciate the thoughtful and constructive feedback provided by the reviewer. We are excited to hear your opinion that our method and experiments are intuitive and that our analysis is comprehensive. These appraisals also seemed to be shared by other reviewers. We are also very grateful for your suggestions to further improve the paper. We will address your major concern below.
>
> **Inter-group interactions.** In our opinion, this is an excellent suggestion. Indeed with our current formulation, we are not accounting for inter-group interactions. An interesting direction for future work would be simplifying group-based representations and using possibly graph-based models to model inter-group interactions. This may also speed up our model’s computation time. Inter-group interaction is a higher level of crowd behavior than grouping. This means that meaningful inter-group interactions might be rare occurrences, so we will need a larger and useful dataset to work with. We have added this as an extension in our discussion of future works.
>
> **Additional points.** Indeed we misused the word “autoencoder”, because this would also mean that we need to change the labels of our approaches (group-auto and laser-group-auto), we decide not to implement the revisions for the rebuttal round in order to not cause confusions among the other reviewers. We will fix the wording “autoencoder” for the camera-ready version.
>
> We added additional details in the appendix to explain the footnote. We also added the work by Kiss et al. Thank you for identifying these typos and suggesting these improvements.

---

### Official Review · Reviewer_qTAW · 2021-07-27

**Originality:** Very Good
**Technical Quality:** Very Good
**Clarity Of Presentation:** Excellent
**Impact:** 4

**Recommendation:**

Strong Accept: I recommend accepting the paper and will argue for my recommendation even if other reviewers hold a different opinion.

**Summary:**

The authors seek to demonstrate the benefits of modeling group structures in social navigation settings.

Their results clearly demonstrate the benefits of modeling groups over individuals for social navigation problems. They also show that their method is also robust to sensor noise.

**Issues:**

See above
1. Experiments on a real robot
2. More analysis of group-nopred vs group-auto

**Reviewer Expertise:**

Very good: Comprehensive knowledge of the area

**Strengths And Weaknesses:**

The paper is clear and well written. It is easy to understand the contributions of the authors, and they clearly demonstrate their contributions.

I appreciated the way the authors clearly laid out their hypothesis (H1-H4) in the evaluation section and then clearly laid out their analysis with respect to those same hypothesis(H1-H4). That organization style made it very easy to understand the questions they were asking, and immediately put the results into context.

Another hypothesis to explore that would be of interest would be comparing group-nopred and group-auto. This would directly highlight the impact of modeling future group motion using the auto encoder. I would hypothesize that group-auto would be “better across the board”: e.g. higher success rate, maintain a higher min-dist-to-pedestrian, have fewer trials with group intrusion, shorter path length. The charts seem to mostly suggest that this is the case, but there are some cases where group-nopred is “better” (e.g. Online, UNIV-F, Success rate). It would be nice to get an in depth explanation of some of these cases and what is going on there.

It would have been nice to see a sim2real deployment. It would have been nice to see how this system behaves in a couple real human scenarios where the environment is much less controlled than this simulation. It would also be nice to understand what are the main limitations of transferring this system to a real robot (i.e. compute bottlenecks, sensor failure modes, etc.)

**Summary Of Recommendation:**

I think the contribution in this paper is clearly demonstrated and with a with a bit more analysis (see above), it could be strengthened.

---

> ### Author Response · Authors · 2021-08-28
> **Author Response to qTAW**
>
> We greatly appreciate the thoughtful and constructive feedback provided by the reviewer. We are excited to hear your opinion that our presentation is clear and that our method and experiments are intuitive. These appraisals also seemed to be shared by other reviewers. We are also very grateful for your suggestions to further improve the paper. We will address your major concerns below.
>
> **Group-nopred vs group-auto.** This is a very good suggestion. We performed analysis between group-nopred and group-auto. Indeed, if we focus on the means, group-auto is “better across the board” in most scenarios, albeit not always statistically significantly better. However, as you mentioned, in a few scenarios, group-nopred is actually “better” than group-auto (UNIV-F success rate is actually higher for group-auto in the online scenario, maybe you meant offline?). In these scenarios group-nopred is not statistically significantly better either.
>
> Two of the reasons that we think are behind this issue are prediction accuracy and the freezing robot problem. The prediction accuracy is simply how accurate our future group predictions are. Sometimes, a small prediction error might close a gap that should be there in the future or mislead the robot into an area that will in fact be occupied in the future. As group-based representations in general occupy a greater obstacle space when compared to individual-based representations, such prediction errors are more likely to misguide the robot into committing unsafe or inefficient navigational actions. As for the freezing robot problem, although group-auto is not statistically worse than group-nopred, the robot does behave more conservatively in group-auto. This means that when surrounded by pedestrians, the robot in group-auto is more likely to stay in place or move back and forth because it is being more conservative in estimating the dynamics of the surrounding pedestrians. This is reflected in lower efficiency and more timeout failures.
>
> We have included this discussion as an additional paragraph in the discussion of our results. As mentioned in the discussion of future work, we are working on improving both the prediction model and the MPC controller towards further enhancing our approach.
>
> **Sim2real deployment.** Due to complications related to Covid-19, we were unable to perform experiments in the real world as this would require participants walking in groups and thus the study would likely involve a certain population density to take place. Unfortunately, this will violate the social distancing guidelines and put the participants at risk. However, we did try our best to perform an in-depth empirical analysis in simulation, which we see as the first step towards a real-world deployment in the near future. In our offline mode, we played back trajectories of pedestrians from real-world datasets to simulate real world pedestrian behavior. While in such a setting our robot is effectively invisible (simulated pedestrians cannot react to it), this assumption is sometimes appropriate, e.g., if the robot has a small unnoticeable profile, or that all the pedestrians around the robot are in a hurry and do not want to give way to the robot. In our online mode, we used ORCA as a model to simulate reactive pedestrians. ORCA has been used in other popular social navigation model evaluations [8, 57, 58]. Lastly, we introduced simulated 2D lidar scans and injected noise into the simulated scan points to emulate aspects of real-world deployment.
>
> We do acknowledge that implementing group-based representations into the real-world presents several (sometimes unforeseen) challenges. We acknowledge that our implementation of group-based representations has not yet been optimized. However, we would like to point out that this paper’s focus is on exploring the utility of group-based representations and we put our planned optimizations as future work. We are working on a newer iteration of group-based representations to improve computation time. This includes simplifying group-based representation geometry and predicting future group states directly in metric space. We only briefly mentioned this in our paper in the discussion section and we have expanded it as part of our future works to make our model more applicable.

---

### Official Review · Reviewer_6KBt · 2021-08-02

**Originality:** Good
**Technical Quality:** Very Good
**Clarity Of Presentation:** Good
**Impact:** 3

**Recommendation:**

Weak Accept: I recommend accepting the paper, but will not argue for my recommendation if the majority of other reviewers have a different opinion.

**Summary:**

This paper tackles the problem of socially compliant navigation. Specifically, it clusters the pedestrians into groups, finds the convex hull of each group, and then included in the cost of an MPC controller for the robot. The method is validated to achieve better safety and somewhat better "comfort" on simulated scenarios, with injected noise.

**Issues:**

Need clarification on the simplicity of the robot dynamics. Discussion or ideally empirical evaluations on how much considering social groups add, instead of just having a controller that can be evaluated fast.

**Reviewer Expertise:**

Very good: Comprehensive knowledge of the area

**Strengths And Weaknesses:**

Overall, the propoed method is intuitive and sound. The method and its motivation is fairly intuitive: we want the robot to take into account the group formation it is in, for more socially compliant navigation.

A main weakness is the simplicity of the robot model that is compatible. Lines 165-169 indicate that the method requires a fairly smalll number of discrete actions to be made. This suggests that in practice the proposed method is too slow, i.e. we cannot solve for continuous steering angle/ velocity. This begs the question, is it worthwhile to consider group dynamics? Would it be better to instead have a MPC controller running at high frequencies and adapt to simpler predictive models. It would be good if this could be investigated in the experiments -- it is unclear whether the MPC controllers used as comparison are also restricted to discrete actions, and are running at the same frequency.

Additionally, there are other methods that merit mentioning. Particularly, POMDP-based solutions DESPOT and its variants (HyP-DESPOT) (Adhiraj Somani et al., 2013, neurips; Panpan Cai et al., 2021 IJRR); and other control-based social navigation approaches (Weiming Zhi et al., 2021, ICRA).

Despite the concern about the simplicity of the robot model, the proposed method is fairly novel and solid. The paper is also accompanied by sufficient empirical evaluation.

**Summary Of Recommendation:**

See above

---

> ### Author Response · Authors · 2021-08-28
> **Author Response to Reviewer 6KBt**
>
> We greatly appreciate the thoughtful and constructive feedback provided by the reviewer. We are excited to hear your opinion that our method is novel and intuitive and that our empirical evaluation is sufficient. These appraisals also seemed to be shared by other reviewers. We are also very grateful for your suggestions to further improve the paper. We will address your major concerns below.
>
> **Robot model simplicity.** We would like to point out that as the idea of incorporating group-based representations into planning/control is very recent, this paper focuses on investigating its utility for a standard MPC architecture that considers discrete actions. Our experiments are structured in a way to show that given the same MPC controller, group-based representations provide benefits in terms of safety and fewer group intrusions. However, alternative MPC architectures could be employed and our general framework is not constrained to discrete actions; with small modifications, one could reformulate our MPC as a continuous optimization problem.
>
> That being said, your question on the computational footprint of our architecture is very relevant. If deploying group-based representations are more computationally expensive, wouldn’t it be better to use a simpler (say an individual-based representation) but have the controller run at a higher frequency? In this paper we focus on investigating the utility of group-based representations assuming that there is a way to use them efficiently. Thus, we did not put much scrutiny into optimizing the computational implementation. Indeed, our current implementation of group-based representations is computationally expensive, but we believe group-based representations can be more computationally efficient than using individual-based representations with modest engineering effort. We have identified certain improvements to our current implementation of group-based representations, including simplifying group shape geometry and exploring models that predict future group states in the metric space so that there won’t be any conversion between metric space and image space. The computation time improvement was very briefly covered as future works in the discussion section. We have expanded this part and suggested that we will improve computation time to improve the applicability of our approach.
>
> Finally, from your comments, we realize that we did not include sufficient details on our MPC controller in the experimental settings. We would like to clarify that for all our baseline policies and our approach, the exact same MPC controller is used. This includes the exact same set of candidate discrete actions as mentioned at the end of section 5 and more importantly, the exact same frequency at 10Hz. To ensure a fair comparison, the same MPC controller is used for our baselines incorporating individual-based representations. We have added additional sentences in section 6.1 (Experimental Setup) when presenting the baseline policies to clarify that the same MPC controller is used.
>
> **Additional literature.** We greatly thank you for the literature suggestions. We have reviewed these works and included all of them in our related works section.

---

### Meta-Review · Area_Chair_B7sX · 2021-08-14

**Recommendation:** Accept (Oral)
**Confidence:** 4

**Metareview:**

Summary: This paper proposes a method for modeling pedestrian agents in groups as opposed to as individuals. The proposed technique designates the groups using a neural network, and the groups’ motion is used within an MPC framework. The proposed method is supported by simulation results.

Originality: The paper presents an interesting and original idea. Drawing inspiration from psychology, the paper proposes a macroscopic view of the world and show that this reduced fidelity can have benefits to improved navigation behavior. Reviewers commented positively on the appeal behind this idea.

Clarity: The paper is very clear in presenting the problem, the intuition and technical approach to the solution, and organizing the analysis of the experimental results. Reviewers seemed generally convinced and interest in the idea of using group-based reasoning to induce socially-aware navigation.

Quality: Reviewers commented on the thoughtful set-up of the experiments to approximate real-world conditions through using public datasets. However, a number of reviewers gave suggestions to further improve the experimental analysis.

Significance: Reviewers generally saw the potential for impact for this work, but were mixed on the applicability of the proposed technique to real-world scenarios. (See “opportunities for improvement” for more details.)

Pros:
This work presents a compelling and novel framing of the navigation problem, a clearly described solution, and thoughtful analysis of experiments.

Main areas of improvement:

Reviewers broadly responded very positively to this paper. The major source of mixed feedback was whether or not the paper did enough to show that the proposed approach is practical. In particular, Reviewers rGtm, qTAW, and 6KBt suggest more discussion or simulation experiments to give insight into what it would take to deploy this technique on a real robot. I would suggest real-work applicability as an area of focus in the revision and rebuttal.

Thank you for considering our feedback, and we look forward to seeing the updated paper.

================
Final Decision

This paper seems like a clear accept - all reviewers recommended accept, even prior to revisions. Where I am not 100% certain is the boundary between oral v poster. I ultimately recommended oral because two reviewers recommended strong accept, all reviewers gave very good or above scores for technical quality, 3/4 gave very good scores for originality, and other scores were positive by consensus.

---

> ### Author Response · Authors · 2021-08-28
> **Author Response to Area Chair**
>
> We would like to thank the meta-reviewer for organizing the reviews and summarizing the suggestions. We are excited to see that the reviewers are uniformly positive about the novelty and intuitiveness of our approach, the thoughtfulness of our experiments and the clarity of our presentation.
>
> The major point of concern appears to be the real-world applicability of our approach. We would like to emphasize that this paper is a proof-of-concept paper that demonstrates the utility of incorporating our group-based representations. As mentioned in the future work section, many optimizations can be made to our current approach. One key area that we plan to focus on is improving the computational efficiency of our approach. We briefly addressed this topic at the end of the discussion section. Finally, we would like to point out that despite not including real-world experiments, our in-depth simulation study involving both real-world data and ORCA-driven crowd environments enables us to make an important first step towards the goal of robot deployment.
>
> We made clarifications to 6KBt to address their concern of whether we are using the same MPC controller across all the policies. We also mentioned our planned improvements to our MPC with group-based representations to bring its computation time lower than MPC with individual-based representations. We included the suggested literature provided by 6KBt.
>
> We took qTAW’s suggestion and added a paragraph in section 6.2 to address group-nopred vs group-auto. We again brought up our planned improvements to G-MPC to reduce its computation time for better applicability.
>
> We added inter-group interaction as possible future work as suggested by UWWw and addressed the minor points mentioned by the reviewer.
>
> Finally, we explained the rationale behind our simulation experiments to rGtm and we believe that the combination of real-world data and crowd simulators enabled us to approach realistic grouping behaviors in virtual settings.

---

### Decision · Program_Chairs · 2021-09-13

**Decision:**

Accept (Oral)

**Comment:**

Summary: This paper proposes a method for modeling pedestrian agents in groups as opposed to as individuals. The proposed technique designates the groups using a neural network, and the groups’ motion is used within an MPC framework. The proposed method is supported by simulation results.

Originality: The paper presents an interesting and original idea. Drawing inspiration from psychology, the paper proposes a macroscopic view of the world and show that this reduced fidelity can have benefits to improved navigation behavior. Reviewers commented positively on the appeal behind this idea.

Clarity: The paper is very clear in presenting the problem, the intuition and technical approach to the solution, and organizing the analysis of the experimental results. Reviewers seemed generally convinced and interest in the idea of using group-based reasoning to induce socially-aware navigation.

Quality: Reviewers commented on the thoughtful set-up of the experiments to approximate real-world conditions through using public datasets. However, a number of reviewers gave suggestions to further improve the experimental analysis.

Significance: Reviewers generally saw the potential for impact for this work, but were mixed on the applicability of the proposed technique to real-world scenarios. (See “opportunities for improvement” for more details.)

Pros:
This work presents a compelling and novel framing of the navigation problem, a clearly described solution, and thoughtful analysis of experiments.

Main areas of improvement:

Reviewers broadly responded very positively to this paper. The major source of mixed feedback was whether or not the paper did enough to show that the proposed approach is practical. In particular, Reviewers rGtm, qTAW, and 6KBt suggest more discussion or simulation experiments to give insight into what it would take to deploy this technique on a real robot. I would suggest real-work applicability as an area of focus in the revision and rebuttal.

Thank you for considering our feedback, and we look forward to seeing the updated paper.

================
Final Decision

This paper seems like a clear accept - all reviewers recommended accept, even prior to revisions. Where I am not 100% certain is the boundary between oral v poster. I ultimately recommended oral because two reviewers recommended strong accept, all reviewers gave very good or above scores for technical quality, 3/4 gave very good scores for originality, and other scores were positive by consensus.